# Temporal nanofluid environments induce prebiotic condensation in water

Andrea Greiner de Herrera[1,2,3], Thomas Markert [4] & Frank Trixler [1,3,5] ✉

Water is a problem in understanding chemical evolution towards life's origins on Earth. Although all known life is being based on water key prebiotic reactions are inhibited by it. The prebiotic plausibility of current strategies to circumvent this paradox is questionable regarding the principle that evolution builds on existing pathways. Here, we report a straightforward way to overcome the water paradox in line with evolutionary conservatism. By utilising a molecular deposition method as a physicochemical probe, we uncovered a synergy between biomolecule assembly and temporal nanofluid conditions that emerge within transient nanoconfinements of water between suspended particles. Results from fluorometry, quantitative PCR, melting curve analysis, gel electrophoresis and computational modelling reveal that such conditions induce nonenzymatic polymerisation of nucleotides and promote basic cooperation between nucleotides and amino acids for RNA formation. Aqueous particle suspensions are a geochemical ubiquitous and thus prebiotic highly plausible setting. Harnessing nanofluid conditions in this setting for prebiotic syntheses is consistent with evolutionary conservatism, as living cells also work with temporal nanoconfined water for biosynthesis. Our findings add key insights required to understand the transition from geochemistry to biochemistry and open up systematic pathways to water-based green chemistry approaches in materials science and nanotechnology.

[1] Department of Earth and Environmental Sciences, Ludwig-Maximilians-Universität München, Theresienstraße 41, 80333 Munich, Germany. [2] Center for Neuropathology and Prion Research (ZNP), Ludwig-Maximilians-Universität München, Feodor-Lynen-Str. 23, 81377 Munich, Germany. [3] School of Education, Technical University of Munich and Deutsches Museum, Museumsinsel 1, 80538 Munich, Germany. [4] Institute of Theoretical Chemistry, Ulm University, Albert-Einstein-Allee 11, 89081 Ulm, Germany. [5] Center for NanoScience (CeNS), Ludwig-Maximilians-Universität München, Schellingtr. 4, 80799 Munich, Germany. ✉email: trixler@lrz.uni-muenchen.de

Prebiotic chemistry is facing a serious problem in regard to the role of water in the emergence of life on Earth: although water is essential for all life as we know it, key biochemical reactions such as the polymerisation of nucleotides into ribonucleic acid (RNA) are impeded in watery solutions[1]. In aiming to overcome this so-called "water problem" in prebiotic chemistry[2], several hypotheses have been proposed[3–6]. Among them are concepts of adding condensing agents such as cyanamide, using alternative solvents such as formamide, setting high temperatures of about 160 °C, or designing prebiotic scenarios based on wet/dry cycles. However, when appraising the plausibility of such concepts and scenarios, general weaknesses appear[4]. A key aspect is that all known life manages the water problem within a stable environment full of water and does not rely on physical conditions and chemical substances proposed in these concepts. This aspect is of particular relevance regarding evolutionary conservatism—the principle that evolution builds on existing pathways[7]. In this context, the principle indicates that the same physicochemical effects were involved in the abiotic origin of biopolymers, as is now being tapped by living systems via complex enzymes.

Living cells contain an intracellular aqueous fluid that is crowded with large, complex biomolecules. In this environment, virtually all water exists as interfacial water[8]. When viewing this dense mixture from the perspective of materials science, it is describable as an aqueous suspension of highly concentrated nanoparticles. In the vicinity of such particles, various nanofluid phenomena emerge in interfacial and nanoconfined water[9, 10]. Consequently, such water differs significantly compared to bulk water in terms of properties such as flow behaviour, reactivity, H-bonding network dynamics, density, dielectric constant, or the quantum state of protons[11–14]. From a geochemical point of view, aqueous suspensions of mineral particles of micro- and nanoscopic size can be regarded as a comparable environment that generates nanofluid effects.

Inspired by this similarity, we designed experiments to test the potential of nanofluid environments within aqueous suspensions of particles for inducing key biochemical reactions in a possibly prebiotic context. As various prebiotic synthesis pathways of nucleosides[15] and nucleotides[4] have been proposed and because the abiotic condensation of nucleotides into RNA within water is a common goal of prebiotic chemistry[16, 17], we chose the polymerisation of nucleotides into RNA as an example reaction. The focus was set on the formation of a pure adenosine-based RNA (poly(A) RNA) since adenosine monophosphate (AMP) is the most common nucleotide in living cells[18]. Furthermore, poly(A) RNAs are common RNAs in cells in the form of poly(A)-tails of messenger RNAs during the process of protein biosynthesis. Thus, we first prepared samples that contained dissolved AMP. To create a nanofluid environment for AMP solutions, we selected quinacridone (QAC) as an example of polyaromatic heterocycle particles and graphite as an inorganic suspended particle species, as both compounds have been well characterised in terms of inducing nanofluid phenomena in aqueous suspensions[19].

According to our results on fluorometric RNA quantification, reverse transcription quantitative polymerase chain reaction (RT-qPCR), melting curve analysis, and capillary gel electrophoresis, abiotic polymerisation of nucleotides into RNA can occur within aqueous dispersions of polyaromatic hydrocarbons and inorganic minerals in the absence of enzymes or additives. We present the synthesis of various amounts of RNA depending on variations of the experimental parameters and propose an explanatory model based on nonclassical effects on water when temporarily confined by suspended particles. The model describes how such temporal, fluctuating nanoconfinements can solve the water paradox by altering crucial physicochemical properties of water and how evolutionary conservatism can link these dynamic natural nanofluid reaction vessels to enzymatic biopolymerisation in biological cells.

## Results and discussion

**Fluorometric poly(A) RNA quantification.** Figure 1a shows the results of fluorometric RNA concentration measurements of such samples. Both ethanol precipitation and isopropanol precipitation

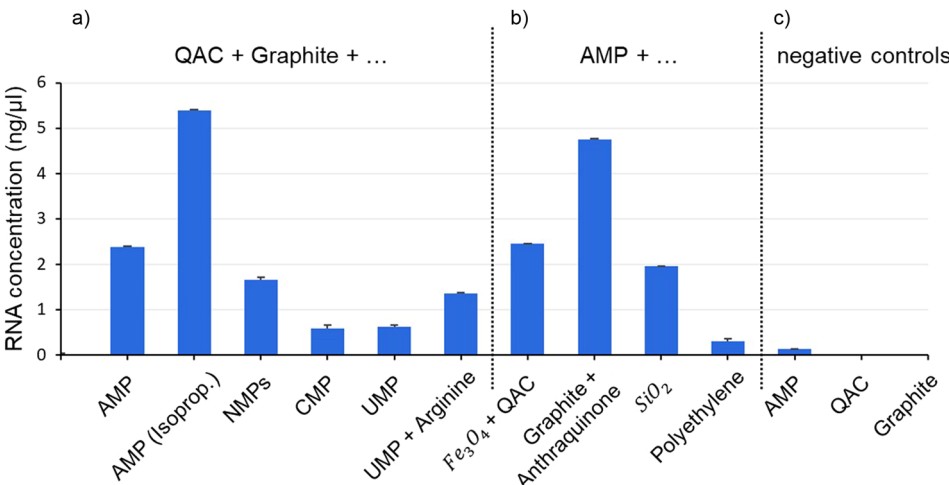

**Fig. 1 Fluorometric results of RNA concentration measurements in aqueous particle suspensions.** Displayed are the mean values of three measurement replicates and standard deviations. If not differently indicated, nucleotide suspensions had a final concentration of 50 mM and RNA isolation was performed from aqueous suspensions of particles containing dissolved biomolecules using EthOH and NaCl for precipitation. **a** Detected RNA concentrations in samples based on QAC + graphite suspensions: AMP: 2.38 ng per µl; AMP (isoprop.): precipitation was performed using isopropanol and NaOAc; detected concentration: 5.4 ng per µl; mixture of NMPs (AMP, UMP, GMP, CMP, 12.5 mM each): 1.66 ng per µl; CMP: 0.59 ng per µl; UMP: 0.62 ng per µl; mixture of UMP and arginine: 1.36 ng per µl. **b** Detected poly(A) RNA concentrations in suspensions based on: Fe3O4/QAC: 2.45 ng per µl; graphite/anthraquinone: 4.76 ng per µl; SiO₂: 1.96 ng per µl; polyethylene: 0.3 ng per µl. **c** Results of the negative controls: aqueous samples containing AMP but without suspended particles: 0.14 ng per µl; pure aqueous samples only containing QAC or graphite: both underneath the detection level.

were applied. The results of both precipitation methods reveal that poly(A) RNA strands have formed in significant amounts regarding the negative controls (Fig. 1c). After having incubated a sample for over 4 weeks at room temperature, the presence of poly(A) RNA was still detected in a concentration (2.53 ng per μl) similar to a comparable suspension sample to which the standard incubation protocol was applied (2.38 ng per μl, Fig. 1a). Negative controls had been performed with all substrate and organic pigment powders used in this study. None of those powders led to a detectable RNA concentration. Furthermore, the RNA concentration of aqueous solutions of mononucleotides was measured, resulting in no detectable RNA signal in the Qubit® Fluorometer. Therefore, RNA contamination of the educts can be excluded. Furthermore, a negative sample using only AMP was analysed to find out if poly-A-RNA is also formed by incubation or precipitation when no substrate or organic pigment is present. The concentration of this sample is lower than the level of RNA amount, reproducibly quantified by the manufacturer of the Qubit® Assay, and can therefore be taken as lower than the reliable detection level, though it seems that very small amounts of poly-A-RNA are also formed without adding substrates or organic pigments.

**Quantitative RT-qPCR.** To cross-check the fluorometric quantification with a totally independent but also highly specific method, we performed quantitative polymerase chain reaction after reverse transcription (RT-qPCR). The results (Fig. 2a) show that the signal of the sample exceeds the background fluorescence after 7.57 cycles. This indicates a high yield (103-fold) of input RNA in comparison to the respective signal of the positive control miRNA (14.25 cycles). To obtain indications on the length of the formed RNA, we performed a comparative melting curve analysis after the last cycle of the qPCR. This analysis included the sample RNA and a positive control miRNA with a length of 23 nt. The comparison reveals that the medium melting point of the sample

is at a higher temperature than the positive control (Fig. 2b). These results suggest that the sample contains created oligonucleotides with lengths equal to or longer than 23 nt.

**Capillary gel electrophoresis of RNA.** As an additional approach to cross-check the presence of RNA and to acquire more information on the length distribution of any formed RNA strands in our samples, we applied capillary gel electrophoresis by using the Agilent RNA 6000 Nano Kit. We chose this technique due to its high specificity for RNA and—like the other RNA detection techniques we selected for our study—due to its tolerance against residual crystal particles suspended in our samples. For the analysis, we chose a sample that contained QAC and graphite particles suspended in an aqueous solution of AMP and applied EtOH/NaCl for precipitation. The results show a broad distribution of RNA bands in the triplicates of the sample (Fig. 3, lanes 1–3). These clear and numerous bands further support our previous findings that RNA is present in our samples and also give a strong indication not only for the existence of short RNAs lengths below 25 nt but also for the presence of a large fraction of formed RNA strands with lengths of up to 4000 nt and more.

Contamination with any natural RNA can be excluded due to the lack of the prominent rRNA double bands of the smaller and the larger ribosome subunits. To identify possible background noises in gel electrophoresis that might be caused by residual nanoparticles in the samples after precipitation, a negative control with a typical particle concentration was used that represented a sample with no detectable RNA signal from fluorometry. The results (Fig. 3, lanes 4–6) suggest that such possible background noise from residual particles is below the detection limit of the used electrophoresis system. The outcome also indicates that the results of fluorometry and electrophoresis concerning the presence or absence of RNA are consistent.

**Identifying factors that determine abiotically induced polymerisation.** To identify substantial factors that lead to the observed abiotic polymerisation, we first analysed the possible role of π-stacking of nucleoside monophosphates (NMPs), as

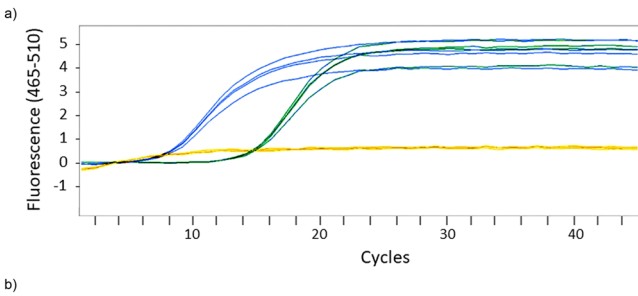

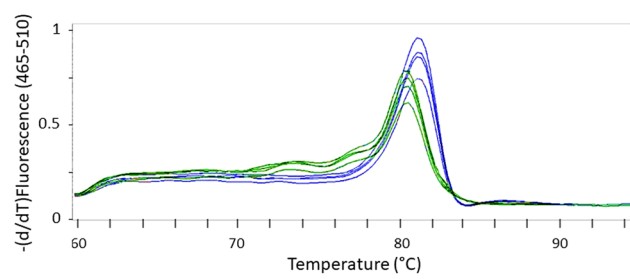

**Fig. 2 Quantitative PCR after reverse transcription (RT-qPCR) applied to an aqueous particle suspension with dissolved AMP.** Yellow: negative control; green: positive control (synthetic miRNA hsa-miR-134-3p (5′-phosphorylated)); blue: sample (precipitation from sample "QAC + Graphite + AMP", Fig. 1a). **a** Fluorescent signals of qPCR. The signals exceed the background fluorescence (yellow) after 7.57 cycles (sample) and 14.25 cycles (positive control). **b** Melting curves. Medium melting point: positive control (green): 80.53 °C, sample (blue): 81.31 °C.

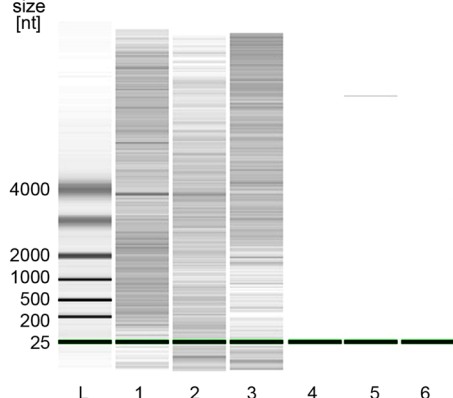

**Fig. 3 Results of RNA gel electrophoresis.** Lane L shows the RNA bands of the molecular weight marker (ladder). Lanes 1–3 represent the triplicate of a typical incubated, precipitated sample made of suspended QAC and graphite particles and dissolved AMP. This sample was analysed with fluorometry prior to gel electrophoresis and showed an RNA concentration of 2.0 ng per μl. Lanes 4–6 represent triplicates of a negative control sample with no detectable RNA concentration according to fluorometric results. The prominent, isolated sharp signal in lane 5 is interpreted as an artifact, arising from the instrument's sensitivity to even low vibrations of the laboratory bench.

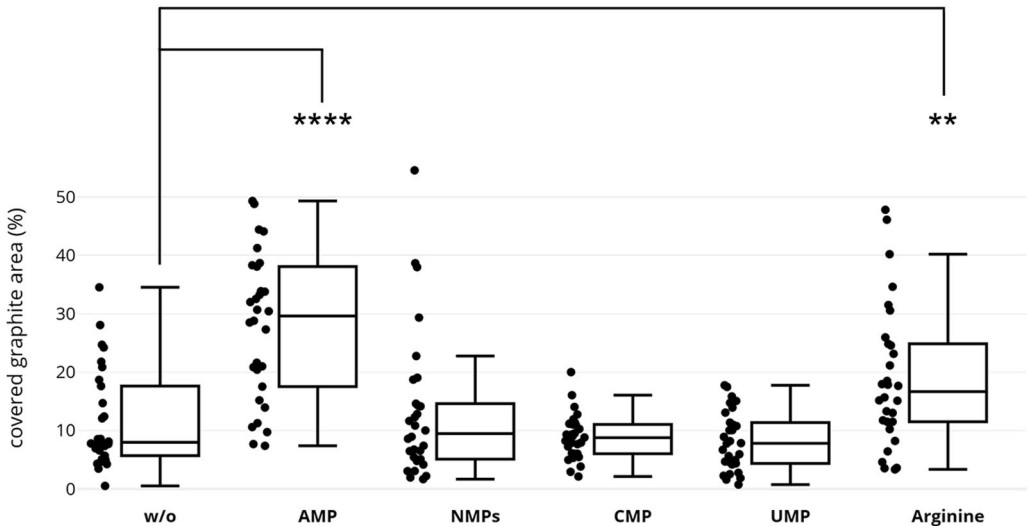

**Fig. 4 OSWD efficiency as a function of biomolecular species added to aqueous particle suspensions.** The box plots and distributions of values are based on results obtained by STM. The statistical analysis of the results reveals the coverage of graphite surfaces with organic semiconductor QAC monolayer. The adsorbate forms after graphite has been brought into contact with a watery suspension of QAC particles that either contains dissolved biomolecules (AMP, NMPs, CMP, UMP, arginine) or no biomolecules (w/o). Testing: *t*-test with Welch's correction; $n = 30$ independent scans; ** $p < 0.01$; **** $p < 0.0001$.

stacking has been suggested to be an important factor for RNA polymerisation in terms of bringing the monomers in close contact[20]. To perform this analysis, we extended the fluorometric analysis of ethanol precipitated AMP-based samples to CMP- and UMP-based samples and compared the relationship of the different RNA concentration results with the relationship of respective published stacking equilibrium constants and stacking free energies of NMPs. The comparison reveals that the detected relative concentration of poly(A) RNA is about four times higher than that of poly(C) RNA and poly(U) RNA (Fig. 1a) and that this order (poly(A) RNA >> poly(C) RNA ~ poly(U) RNA) correlates to the order of the respective stacking equilibrium constants and stacking abilities derived from the stacking free energy profiles of NMPs[21]. This correlation suggests that in particle suspensions, stacking and polymerisation of nucleotides are linked. Therefore, enhancing the stacking ability of a nucleotide should result in a higher RNA concentration. To test this hypothesis, we selected UMP due to its low self-stacking ability and its comparatively low poly(U) RNA formation and added the amino acid arginine to the sample. Arginine is known for its high stacking ability with aromatic groups[22]. Fluorometric analysis of UMP/arginine-based samples revealed that arginine increased the formation of poly(U) RNA by more than 100% (Fig. 1a). This observation supports the necessity of π-stacking for polymerisation in aqueous suspensions and points to a possibly prebiotic relevant kind of interplay between amino acids and nucleic acids.

However, base stacking cannot be the key factor for RNA formation in the described samples. This becomes evident when taking the water paradox into account and when considering the result of a negative control based on an AMP solution without suspended particles (Fig. 1c): although AMP has the highest self-stacking constant among all NMPs[21] and is the only NMP with the ability to self-associate in indefinite stacks[23], the negative control shows that in comparison to the poly(A) RNA concentrations reported above, only minute amounts can be found when the sample contains no added particles. This result suggests that there is a key factor in promoting both stacking and polymerisation that is closely linked to the particle suspension nature of the samples.

It is known that reducing the dielectric constant of water favours stacking[23], and reducing the activity of water (e.g. by adding alternative solvents, inducing wet/dry cycles, or intercalation) promotes polymerisation[3, 24, 25]. As nanofluid phenomena emerging in temporal nanoconfined water can reduce both properties simultaneously[13, 26] and watery suspensions of particles give rise to such phenomena[9], we infer that the occurrence of nanofluid phenomena in our samples is the key factor for the observed nucleotide polymerisation. This implies that the enhancement of nanofluid effects should correlate with an increase in polymerisation reactions.

**Comparing the extent of nanofluid effects to polymerisation efficiency.** Nanoconfining environments change the behaviour of water, especially in terms of its hydrogen bond network dynamics[13], which, in turn, affects the thermodynamic property of water activity[26]. The anomalous behaviour of nanoconfined water results from a highly complex interplay of various nanofluid phenomena and forces that are related to, for example, the surface energy and size of the confining boundaries, shear, molecular structure, electrical double layer, and fluctuations of general order parameter[9, 27]. To cope with this high complexity when assessing a possible synergy between nanofluid phenomena and polymerisation, we chose an experimental approach that allows us to focus on a single, quantifiable effect. This approach uses the effect of organic solid/solid wetting deposition (OSWD) as a probe. OSWD is the final result of a network of various nanofluid phenomena on confined water between suspended organic crystals, including double-layer forces, Casimir-like fluctuation-induced forces, and dewetting-induced hydrophobic collapse[19]. It manifests as the adsorption and self-assembly of insoluble polyaromatic heterocycles at solid/solid interfaces and thus is quantifiable via surface coverage determination.

We quantified the efficiency of OSWD as a function of different biomolecules dissolved in watery suspensions of particles of the organic semiconductor QAC. For quantification, we measured the surface coverage of graphite with QAC monolayers via scanning tunnelling microscopy (see Supplementary Figs. 1 and 2). The results (Fig. 4) reveal that adding AMP to

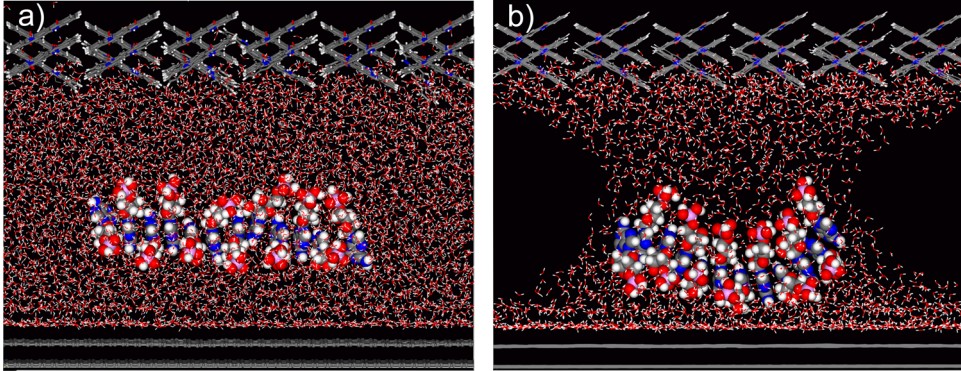

**Fig. 5 Dynamic force field calculation of a nucleotide stack in nanoconfined water.** The system is modelled within a supercell containing two graphene layers (bottom), two layers of a QAC crystal (top) and a stack of 12 AMP molecules placed between the confining surfaces. Vertical dimension of the gap: 43 Å. **a** Condition of the stack after a simulated time span of 50 ps. The confinement is filled with water molecules. **b** Condition of the stack after a simulated time span of 60 ps and a water density of ~60%.

the particle suspension enhances the coverage with very high statistical significance ($p < 0.0001$) in relation to QAC suspensions based on pure water (w/o, Fig. 4), though the pH was not increased to the same extent (pH of aqueous suspension of QAC: 7.5; pH with AMP added: 8.8). A significant increase in coverage ($p < 0.01$) also occurred when using arginine as the added biomolecule (pH 10.6). By contrast, coverages found in samples based on CMP or UMP are statistically nearly identical and significantly lower with respect to the AMP- and arginine-based samples. Using a mixture of different NMPs leads to no significant increase in coverage but results in some very high single coverage measurements regarding the CMP and UMP samples (NMPs, dot plot, Fig. 4), possibly caused by clusters of the AMP fraction of the mixture. The comparison of these OSWD induced coverages with RNA quantities (Fig. 1a) indicates that relative differences in the extent of nanofluid phenomena between the samples correlate with relative differences in detected RNA quantities. This supports the hypothesis of synergy between nanoconfinement effects and nucleotide polymerisation in aqueous particle suspensions. We thus propose that the emergence of nanofluid effects on confined water is the key factor in the promotion of both stacking and polymerisation of nucleotides within aqueous suspensions of particles.

**Assessing the generalisability of suspension-induced polymerisation.** Thus far, we used QAC/graphite particle systems to ensure comparability of the results from different experimental approaches. To test whether the formation of RNA in such an environment is generalisable, we measured samples prepared by the same protocol but containing particles based on other compounds. As inorganic substitutes, we chose magnetite and silica as geologically widespread compounds and selected the organic compound anthraquinone due to its abundance in carbonaceous meteorites[28]. Fluorometric measurements of partly or fully substituted samples show poly(A) RNA concentrations that are similar to or substantially higher than the comparable output of the QAC/graphite model system (Fig. 1b). This indicates that the abiotic formation of RNA within nanofluid environments of aqueous suspensions is of general nature. Our results also imply that the output of RNA can be increased to higher concentrations by identifying appropriate particle suspension systems. The influence of the particle species on the RNA concentration, as indicated by Fig. 1b (including results from inert polyethylene particles as a comparison), is consistent with the fact that the anomalous behaviour of nanoconfined water is partly determined by the characteristics of the confining surfaces[12].

**Dynamic molecular mechanics calculations.** For nanofluid effects to become dominant, the confining surfaces must approach, at a conservative estimate, within the range of 10 nm and below[13]. To evaluate whether stacking of nucleotides within such nanoconfinements is possible in a sufficiently stable and ordered way to prime polymerisation, we performed dynamic molecular mechanics calculations. For performing these calculations, we modelled a stack of 12 AMPs as an example system referring to the observation that polymerised AMPs of more than 10 nt length were abundant in our samples. We arranged this stack in parallel between two nanoscale separated crystals of a QAC/graphite system and added water to the confinement. Figure 5a shows the condition of the stack at the end of a dynamic simulation modelled with a confinement gap size of about 4 nm. This condition implies that the nucleotide stack remained arranged in such confinement when surrounded by water (see also Supplementary Movie 1). Comparative calculations suggest that such stacks are deformed, but stable even when the density of water—which can be lower in nanoconfined conditions[13]—is decreased to ~60% (Fig. 5b and Supplementary Movie 2). It requires a reduction of the gap size to about 3 nm and below to finally destabilise a stack due to increased interactions with confining surfaces (Supplementary Fig. 3). In sum, these simulations indicate that it is feasible to assume that nucleotides can associate in stable stacks within nanoconfinement gap sizes well below 10 nm for priming polymerisation.

**Thermodynamic considerations.** Nucleotide polymerisation into RNA is a water-releasing condensation reaction that faces a thermodynamic barrier as the change in the Gibbs free energy of such a reaction is positive. Consequently, nucleotide polymerisation in water is a thermodynamically uphill reaction that is extremely inefficient to occur spontaneously under ambient conditions. However, if the entropic part ($\Delta S$) of the Gibbs free energy change ($\Delta G = \Delta H - T\,\Delta S$) becomes very positive, the reaction can become exergonic ($\Delta G < 0$) and thus favourable. As the entropic part is positive when the activity of water is low[24], the thermodynamic barrier can be overcome by reducing water activity[29]. Against this background, we propose that our observed abiotic formation of RNA within aqueous particle suspensions can be explained by the rise of anomalous properties of water when getting temporarily confined between suspended particles: nanoscale confinements change, among others, the vapour pressure[30] and hydrogen-bond network dynamics[13] of water. This can reduce water activity[26, 30] as a function of gap size and the characteristics of the confining surfaces. We suggest that in

comparison with other ways to circumvent the thermodynamic barrier of such condensation reactions, the exergonic impact of nanofluid phenomena in aqueous particle suspensions on RNA polymerisation and stabilisation is of high relevance for prebiotic plausibility, as it does not require nonphysiological conditions such as temperatures well above 100 °C, alternative solvents, or wet/dry cycles.

## Conclusion

Our results indicate that abiotic temporal nanoconfinements of water within aqueous particle suspensions serve as nanoscopic, flexible reaction vessels for prebiotic, nonenzymatic RNA formation. The findings can solve the water paradox in such a way that nanofluid effects in aqueous particle suspensions open up an abiotic route to biopolymerisation and polymer stabilisation under chemical and thermodynamic conditions that are also prevalent within the crowded intracellular environment of living cells. The fact that polymerase enzymes also form temporal nanoconfined water clusters inside their active site[31, 32] implies that the same physicochemical effects are tapped for nucleotide condensation in water both by biochemical pathways and our reported abiotic route. These aspects indicate that our model is consistent with evolutionary conservatism stretching back to the era of prebiotic chemical evolution and the origin of cellular life[33]. The consistency is further supported by the fact that water is not trapped by nanoconfinements within the polymerase core but can exchange with the surrounding intracellular fluid[32]—a situation that also exists in abiotic nanoconfinements of water emerging temporarily between approaching crystal particles in aqueous suspensions. Our experimental finding that under these conditions an amino acid catalyses the abiotic polymerisation of nucleotides supports first indications of the role of minerals for the origin of cooperation between amino acids and nucleotides[34] evolved to the interdependent synthesis of proteins and nucleic acids in living cells[35]. Abiotic RNA polymerisation under nanofluid conditions in aqueous particle suspensions does not depend on specific mineralogical and geological environments: now as then, in the prebiotic world, watery suspensions of micro- and nanoparticles are virtually ubiquitous[36]. They exist, for example, in the form of sediments with pore water[37], hydrothermal vent fluids containing precipitated inorganic[38, 39] and polyaromatic[40] particles, atmospheric aerosols with organic and inorganic particle burden[41, 42], or crowded, dispersed aggregates inside water-filled cracks in the crust of the earth[43, 44], and possibly of icy moons such as Enceladus[45].

Beyond the discussed importance of our findings for prebiotic chemistry and theories on the transition from geochemistry to biochemistry in the course of life's emergence, the results are also highly relevant for the development of sustainable chemistry approaches in applied sciences: the detected substantial efficiency increase of OSWD induced monolayer formation of widely used industrial organic semiconductors such as QAC (Fig. 4) or phthalocyanine[46, 47] (see Supplementary Fig. 4) via biomolecules with a prominent stacking capability such as AMP and arginine provides a systematic path to catalyse the deposition of water-insoluble organic semiconductors on 2D materials via nanofluid water. This insight is substantial for the development of green chemistry approaches to chemical surface engineering, band-gap engineering, and low-dimensional crystal engineering in materials science and nanotechnology.

## Methods

**Compounds**. Adenosine-5′-monophosphate (AMP, Sigma, disodium salt, CAS-Nr. 61-19-8); Uridine-5′-monophosphate (UMP, Alfa Aesar, disodium salt, CAS-Nr. 58-97-9); Cytidine-5′-monophosphate (CMP, Alfa Aesar, disodium salt, CAS-Nr. 6757-06-8); Guanosine-5′-monophosphate (GMP, Alfa Aesar, disodium salt, CAS-Nr. 85-32-5); L-Arginine (Alfa Aesar, CAS-Nr. 74-79-3); graphite (HOPG, purchased from NT-MDT, item no. GRBS/1.0); Quinacridone (average primary particle size: 70 nm, Clariant GmbH, CAS-Nr. 1047-16-1); Anthraquinone (purity 97%, Fluka, CAS-Nr. 84-65-1); Phthalocyanine (β-form, purity 98%, Sigma-Aldrich, CAS-Nr. 574-93-6); carbon nanopowder (<50 nm (TEM), purity ≥ 99%, Sigma-Aldrich, CAS-Nr. 7440-44-0); carbon (mesoporous, 0.25 cm³/g pore volume, Sigma-Aldrich, CAS-Nr. 1333-86-4); Iron (II, III) oxide (magnetite, 50–100 nm particle size, purity 97%, Sigma-Aldrich, CAS-Nr. 1317-61-9); Silicon dioxide (-325mesh, purity 99.5%, Sigma-Aldrich, CAS-Nr. 60676-86-0); Polyethylene (<400 micron, Alfa Aesar, CAS-Nr. 9002-88-4).

**Suspension preparation**. Three millilitre of aqueous suspension was made of each 0.1 g/ml inorganic substrate (e.g., graphite powder) and/or organic pigment with a total nucleoside monophosphate concentration of 50 mM. Samples were incubated overnight at 60 °C while mixed horizontally at 300 rpm, to avoid sedimentation of substrates and pigment. After incubation, samples were centrifuged at 8000 × g for 1 min at room temperature (RT). The supernatant was transferred to a new collection tube.

**Precipitation**. If not differently indicated, precipitation was carried out with 0.2 M NaCl and 3.5 volumes of EtOH. One sample was precipitated with 0.3 M NaOAc (pH 5.2) and 0.7 volumes of isopropanol to test the efficacy of a different precipitation method. After the addition of the appropriate amount of salt solution and alcohol, samples were mixed by inverting the tubes 5 times, and precipitation reactions were incubated 24 h at −20 °C. Afterwards the samples were subsequently centrifuged at 14,000 × g, for 1 h without cooling (room temperature between 20 and 25 °C). The supernatant was discarded, leaving ~20 µl of it inside the reaction tube, in addition to any formed gel pellet. Formed gel pellets were dried at 37 °C for 20 min and resuspended in an appropriate amount of nuclease-free water, as little as possible needed to dissolve the gel pellet. Depending on the volume of the gel pellet, more or less volume of water was added. Final RNA concentrations were then normalised to a uniform volume of 200 µl, using the total RNA amount measured and the final volume.

**Fluorometric RNA quantification**. Concentrations of microRNA suspensions were measured using a Qubit® 3 Fluorometer (Invitrogen™)[47] and the Qubit® microRNA Assay Kit (Invitrogen™) due to its high specificity and reliability: "The assay is highly selective for small RNA over rRNA or large mRNA (>1000 nt) (…), and tolerant of common contaminants such as salts, free nucleotides, solvents, detergents, or protein (…)."[48] This declaration was controlled by measuring a 50 mM aqueous AMP solution with the Qubit® 3 Fluorometer and was validated. Concentrations were calculated by the chosen microRNA or RNA program of the Qubit® 3 Fluorometer. Standard curves and samples were prepared following the manuals[49]. Concentrations were calculated and normalised regarding the respective volume of each sample.

**Quantitative RT-qPCR analysis**. For RT-qPCR, selected samples were reverse transcribed using the TaqMan™ Advanced miRNA cDNA Synthesis Kit (Thermo Fisher) following the manual. As a positive control, the synthetic miRNA hsa-miR-134-3p (5′-phosphorylated) (eurofins) with an oligonucleotide length of 23 nt was used. As a negative control nuclease-free water was used. For qPCR, the Master Mix reaction was composed of 10 µl QuantiTect SYBR® Green (Qiagen), 2 µl amplification primer mix from the TaqMan™ Advanced miRNA cDNA Synthesis Kit (Thermo Fisher), and 8 µl of diluted sample. qPCR, including a melting curve of the formed amplicons after the last cycle, was run on a LightCycler® 480 Instrument II with the following settings: 15 min 95 °C, 40x: 15 s 94 °C—30 s 60 °C —30 s 72 °C—Single Data Acquisition, 72 °C 2 min, Melting curve.

**Capillary gel electrophoresis**. For RNA gel electrophoresis the Agilent RNA 6000 Nano Kit was used and the gel setting was prepared following the manual. Samples were run on a 2100 Bioanalyzer Instrument with the 2100 Expert Software (Agilent). Samples were applied in triplicates. As a negative control a particle suspension based sample was used that resulted in a "too low" signal in the Qubit® concentration measurement, to identify possible background noises that might have been generated by residual nanoparticles leftover in the samples after precipitation, like those of inorganic substrates or organic pigments.

**Scanning tunnelling microscopy**. For scanning tunnelling microscopy (STM) investigations aqueous suspensions were prepared of each 2% w/w quinacridone pigment and a nucleotide monophosphate or amino acid final total concentration of 25 mM, if required. Suspensions were drop cast onto graphite substrates to induce OSWD and dried overnight. The samples were subsequently covered with dodecane to achieve a defined, stable environment between the STM tip and substrate. Each sample was prepared twice. From each replicate, 15 images of 200 nm × 200 nm size were generated from three different spots. Images were acquired with a self-built STM combined with an SPM 100 control system supplied by RHK Technology. The scan settings were: bias = 1 V, tunnel current = 300 pA, and line time = 50 ms. Further, the voltage pulses used to improve the scan quality

were applied within the range of 4.3 V to 10 V. All STM measurements were performed under ambient conditions.

**Surface coverage determination**. To determine coverage, meaning how much area of graphite is covered by supramolecular adsorbate structures (see Supplementary Fig. 1), black/white histograms were created. For this purpose, depicted adsorbate structures were masked out in the STM image (white) while setting the background (graphite without adsorbate structures) to black (see Supplementary Fig. 2). Black/white histogram values were calculated using "histogram" within the "analyse" menu of the software ImageJ. The mean values given in the histogram data box were then converted to percentages of the graphite area covered with adsorbate structures, whereby a histogram value of 0 means 0% of the graphite surface is covered with adsorbate and a histogram value of 255 means 100% of the graphite surface is covered. Note that AMP itself does not form monolayer on graphite due to its 3D-structure. For statistical analysis of the coverage results, a $t$-test with Welch's corrections was used since the samples show unequal distribution variance. The calculations were performed using Excel, giving the results of a two sided unequal variance $t$-test.

**Computer simulations**. All molecular mechanics calculations were performed with the Materials Studio package (Accelrys). The force field used was Dreiding[50], where partial charges of atoms within the molecule are calculated with the Gasteiger method[51]. Geometry optimisation calculations were calculated where convergence tolerance regarding energy was $1.0e-4$ kcal/mol, force was 0.005 kcal/mol/Å and displacement was $5.0e-5$ Å. The Smart algorithm was used which is a concatenation of steepest descent, Newton-Raphson and quasi-Newton methods, to obtain better behaviour for the different stages of downstream minimisation. Dynamic calculations were performed for an NVE ensemble with a temperature of 298 K, random values assigned for the initial velocities of the atoms and a time step of 0.1 fs for the integration algorithm.

A chain of 12 AMP molecules was constructed starting with building an AMP dimer. The dimer itself was built by duplicating an AMP and performing an 180° rotation related to the axis lying in the plane of the adenine molecule and passing through the centres of the hexagon and pentagon. Afterwards, the rotated AMP was translated normal to the plane of the adenine by a distance of 3.5 Å. This dimer was stacked 6 times in a row such that all planes of the adenine molecules were parallel each by a distance of 3.5 Å. This artificially constructed model was finally optimised related to geometry.

The entire system is modelled within a supercell containing two graphene layers, on top at some distance a chain of AMP molecules and finally two layers of a QAC crystal above the AMP chain. The entire system was filled with water molecules at different concentrations. The initial position of the AMP chain related to the graphene layers and the QAC crystal layers is stated below for the different calculations performed. The vertical dimension of the supercell has been chosen to be safe so that an interaction of the graphene layers and the QAC crystal layers can be neglected (gap is 43 Å). The rectangular in-plane dimensions of the supercell have been determined to minimise the distortion of the periodic structure of the QAC crystal layers. The deviation of the unit cell vectors of the QAC layers is almost zero in one direction (0.1%) but quite large in the other direction (5.6%). However, because the outer layer of the QAC crystal is always kept fixed, the adjacent free movable QAC layer was observed to be stable throughout all simulations. This observation was the reason why this significant deviation was accepted for all calculations. The graphene layers are not distorted with regard to the optimal unit cell dimensions. The dimensions of the supercell are: plane x, y: 80.956 Å, 56.58 Å, and z: 103.4 Å.

## Data availability

Supplementary Information contains: STM images and simulation results on the QAC adsorbate structure; additional dynamic force field calculation results on AMP in nanoconfined water; phthalocyanine related STM results and box plots on AMP catalysed OSWD. Videos of a simulated AMP stack within a confinement filled with water (Supplementary Movie 1, mp4) and with water of a reduced density to ~60% (Supplementary Movie 2, mp4). The datasets generated during and/or analysed during the current study are available from the corresponding author on reasonable request.

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

## Acknowledgements
We thank J. Herms and F. Strübing for technical and practical support regarding qPCR and RNA concentration measurements at the Center for Neuropathology and Prion Research, LMU, Germany, and A. Brachmann for enabling and support the use of the capillary gel electrophoresis system of the Equipment Service, Faculty of Biology, LMU, Germany. F.T. has received funding from the Bavarian State Ministry of the Environment and Consumer Protection grant agreement no. 71k-U8793-2015/14-9. This work was supported by the mentoring program of the Nanosystems Initiative Munich (NIM).

## Author contributions
The manuscript was written through contributions of all authors. All authors have given approval to the final version of the manuscript. A.G.H. designed and performed experiments and data analysis, T.M. designed and performed computer simulations, F.T. conceived the project and directed the research.

## Funding

## Competing interests
The authors declare no competing interests.
