## [Peer review file · Communications Chemistry]

Temporal nanofluid environments induce prebiotic condensation in waterReviewers' comments:

Reviewer #1 (Remarks to the Author):

Title: Temporal nanofluid environments induce prebiotic condensation in water

Dear Authors, dear Editor

I am glad to have had the opportunity to review this manuscript, as the findings that it contains are worth full attention, comment and following in the prebiotic chemistry community. The results in the paper are a novelty and contribute to a substantial advance in the field.

Overview and general recommendation:

One of the most relevant questions in the stepping from chemical synthesis into complex chemistry, which preceded the precellular stage and finally ended with the emergence of biological systems, is the construction of oligomers and polymers from monomers. This process was essential in the complexation of molecules and there is a long debate around this issue. The formation of the main biological polymers involves a condensation reaction involving monomers that always results in the production of water. This reaction is not easily achieved under watery environments, such as those that are commonly presented as likely niches for chemical evolution. This constitutes the paradox of water, in the sense that water is necessary to living beings, despite its quenching role in the emergence of life.

A common purpose in the field is tracing the abiotic synthesis of RNA in water, which includes the following steps: the synthesis of (1) nucleobases, (2) nucleosides and (3) nucleotides, and then (4) the polymerization of nucleotides.

The polymerization of monomers is a big deal. Initial experiments included the use of extreme conditions (pressure, temperature, concentration, etc.) or promoters (catalyzers) that do not occur in Nature. During the last two decades, many studies have been devoted to the synthesis of RNA upon prebiotic conditions, such as the presence of salt, the role minerals, the effect of cycles (dry/wet; heat/cold), etc.

By following the present study by Prof. Trixler and co-workers, the water paradox could be overcome. In this paper the group archives the formation of RNA oligomers (more than 20 nt) in aqueous conditions by exploiting the critical properties that water acquires under confinement in nanofluid conditions.

A key point in the manuscript is the careful experimental design to test their hypothesis. I will describe the main findings and add my comments accordingly.

1) Test the formation of pure adenosine RNA oligomers (polyA RNA), the system included quinacridone (QAC) and graphite (inorganic suspended particle); both (QAC and graphite) are capable of induced nanofluidic phenomena in aqueous suspensions.

Ethanol and isopropanol were used to precipitate polyA oligomers.

They employed two independent methods for measurement of the polymerization, including fluorescence and RT-qPCR technique. Later they made a melting curve.

Comments:

Figure 1 is not clear. Is the "AMP" legend the polyA oligomers <10 nt? And the AMP>10 nt refers to

oligomers with more than 10 nt?

This Figure needs to be clearer, as it resumes the main results obtained in the research. The lines in gray are almost invisible but, as they show different stages in experimental design, some color variation could enhance the visibility of the results.

Figure 2a has too many tags in axis "y", which looks too busy; please omit half of them to make it clearer. For instance, you could omit -0.20, 1.20, 2.60 and so forth. Do not delete the secondary marks.

Figure 2b. Same as before. Please also omit some tags on axis "y", leave the secondary marks.

2) Factors that determine abiotically induced polymerization

n-stacking of nucleoside monophosphates was evaluated as a determining factor for polymerization. Different experiments containing AMP- CMP- and UMP-based samples (+ QAC + graphite) were tested, thus finding a correlation between their data and the published stacking equilibrium constants, and stacking free energies of NMP.

To test the stacking ability to increase the RNA concentration, they analyzed the effect of arginine in UMP-based samples. The production in PolyU yield increased immensely. In this regard, I wonder if the authors made measurements with other nucleobases. I understand that the experiment was designed to test the feasibility to induce a larger polymerization and the selected molecule was UMP, but it could be interesting to test the other NMPs to reinforce the idea of the interaction between amino acids and nucleic acids.

3) Extent of nanofluid effects and polymerization efficiency

The idea was to test the synergy between nanoconfinement effects and nucleotide polymerization in aqueous particle suspensions. They found that the emergence of nanofluidic effects on confined water is the main factor that affects stacking and polymerization.

Comment:

Figure 3. Please explain if the $p < 0.01$ is applicable for all the systems except for the arginine system with $p < 0.0001$. The nomenclature is not clear. The "arginine" sample is only the amino acid, or it also includes the UMP?

4) Generalizability of suspension-induced polymerization

This section is of paramount relevance since it shed light on the process, its generalization and feasibility under prebiotic conditions.

Comments:

The authors designed the experiment with other relevant inorganic materials, like magnetite and silica particles. Please specify the size of such particles and whether they are synthetic or natural solids. If they are natural, did you characterize them by means of geochemical or mineralogical methods? How did you clean the samples and select their size (and why so)?

5) Dynamic molecular modelling

DMM calculations of the nanofluidic properties in the studied systems are appealing. The authors show that the distance between the confining surfaces is key: when the gap is reduced, the nucleotides associate and polymerize. It may be out of my own curiosity, but did you model other surfaces aside from graphite? Considering the great effect that surfaces have on polymerization, as demonstrated in your experiments, could this be the next step?

Further notes

- This manuscript is well written and logically presented; the authors made a review of the main articles published in the fields. I also recommend including this one:

Wu, K., Chen, Z., Li, J., Li, X., Xu, J., & Dong, X. (2017). Wettability effect on nanoconfined water flow. *Proceedings of the National Academy of Sciences*, 114(13), 3358-3363.

<https://www.pnas.org/doi/10.1073/pnas.1612608114>

- With regard to the methods, I have some comments.

Suspension preparation

L. 303. The expression "overnight" is vague, please make explicit the actual time span in hours.

L. 305. The same applies to "Room Temperature (RT)", as is not accurate, please state an actual temperature.

Precipitation

L.316. The phrasing "in an appropriate amount of H₂O dest. " is vague, please explain what do you mean by appropriate amount?

- The statistics of data are robust, but the figures can be improved to make them more attractive and clearer.

After the revision process, I recommend the acceptance of this manuscript with moderate changes, although some key aspects (as stressed above) are in need of further clarification.

Reviewer #2 (Remarks to the Author):

Trixler and coworkers present results from experiments designed to use molecular deposition of mononucleotides on particles in suspension as a means to generate RNA polymers. The authors interpret that their observations as showing polymerization. Even though the monomers and particles are in bulk water, they argue that polymerization is possible, although thermodynamically unfavored, because of the altered properties of water on particle surfaces in what they call "nanoconfinement". The authors further claim that their demonstration of RNA polymer formation in water solves a major challenge to the prebiotic origin of RNA.

This manuscript suffers from two critical flaws: 1) a fallacious rejection of existing models for prebiotic polymerization in favor of their own hypothesis that "nanoconfinement" is a more logical solution to prebiotic mononucleotide polymerization in water, and 2) poorly designed experiments that lack necessary controls, the combination of which results in observations that are misinterpreted by the authors.

Critical Flaw 1): The authors state "Current strategies to circumvent this [the water] paradox have low prebiotic plausibility regarding the principle that evolution builds on existing pathways." This logic, used to reject previous proposals for prebiotic polymerization in an aqueous environment, including the use of chemical condensing agents, alternative solvents, or wet dry cycles, suffers from the fallacy of Extended Analogy. While there are reasons to accept that current metabolic pathways have evolved from earlier pathways with similar reactions, this principle does not mean that every chemical process used in life today was exactly the same during the earliest stages of life. For example, the nucleobases were probably first produced on Earth by reactions that started with HCN, but life today does not use HCN, but amino acids, to build the nucleobases. Furthermore, the authors argue that "nanoconfinement" within enzymes and the different state of water in such restricted spaces justifies their argument that nanoconfinement was key to polymer formation on the early Earth. This too is a fallacious conclusion. The authors seem to be unaware that the free energy stored in pyrophosphate bonds, such as in ATP and GTP, are what drive polymerization in extant life. The microenvironment of an enzyme may protect against pyrophosphate/polymer hydrolysis during a polymerization reaction, but the removal of bulk water is not enough. Polymerization requires the activation energy provided by nucleotide triphosphates.

Critical Flaw 2): 1. The authors describe their RNA polymer product isolation procedure as: "Ethanol precipitation to isolate possible poly(A) RNAs of 2-10 nt lengths and isopropanol precipitation to isolate such RNAs longer than 10 nt were applied. The results of both precipitation methods reveal that poly(A) RNA strands have formed in significant amounts regarding the negative controls (Fig. 1c) and that oligonucleotides longer than 10 nt lengths are abundant in such samples." The authors seem to be unaware of the fact that some AMP will precipitate under the conditions used to precipitate RNA

polymers. The AMP that is precipitated during their described procedure may be giving a false positive reading in their fluorescence assay, and precipitation may be enhanced by their suspended particles. Furthermore, the authors also state: "The supernatant was discarded, leaving approximately 20 μ l of it inside the reaction tube, in addition to any formed gel pellet." A fraction of AMP, if still suspended in ethanol solution, could result in a false positive for product formation in their assays. The authors need to show that no oligomers are present as minor contaminants in their stock supplies of AMP.

Regarding their physical assays, UV-vis melting temperature is used as an assay for poly(A) formation. However, poly(A) does not give a 2-state melt as it would in the presence of a poly(T) complement. Under acidic conditions a hemi-protonated poly(A) duplex is formed, but there is no indication that their conditions would support the formation of this highly pH-dependent structure. Results from their assay shown in Figure 2 reveals that the negative control sample also gives a melting transition above 80°C. The origin of this transition in the control sample is equally perplexing.

The authors use Quantitative PCR after reverse transcription (RT-qPCR) to confirm the synthesis of poly(A) from AMP. This technique is highly prone to false positives due to contamination. It is known that some commercial sources of AMP have poly(A) contaminants. The processes/additives used as part of the RT-qPCR experiments, including particles in suspension, could be facilitating the amplification of such contaminants.

I recommend that authors completely redesign their polymerization assay to include more standard analytical techniques, such as gel electrophoresis, HPLC, and mass spectrometry.

Reviewer #3 (Remarks to the Author):

COMMUNICATIONS CHEMISTRY-REVIEWER

TITLE: Temporal nanofluid environments induce prebiotic condensation in water

Authors: Andrea Greiner de Herrera, Thomas Markert, and Frank Trixler

The life as we know needs water, however, most prebiotic chemistry reactions are not thermodynamically favorable in the presence of it. Thus, this article is about how to get around the water paradox. The paper was well written and the authors presented very good results. In my opinion, this article is interesting for Communications Chemistry readers.

I have few questions that I would like that the authors clarified them.

In the title of the article appears the word "prebiotic" this means that the experiments were performed under reaction conditions that could have existed on the prebiotic Earth.

I agree with the authors that magnetite and silica were widespread minerals on the prebiotic Earth. However, arginine and anthraquinone were not widespread organic compounds on the prebiotic Earth. According Milshteyn et al. (2019) cited by the authors, the anthraquinone concentration in the Murchison meteorite was 2.0 nmoles/g (Sci. Rep. 2019, 9, 12447. DOI: 10.1038/s41598-019-48328-5). This means that the concentration of anthraquinone in the primitive seas of the Earth was very low. Also, arginine is one of the amino acids that were only synthesized in few experiments and always very low amount (Orig. Life Evol. Biosph. 2008, 38: 469–488. DOI 10.1007/s11084-008-9150-5). This issue should be addressed by the authors.

As a suggestion for other works of the group. Could other amino acids be used for these experiments?

Glycine, α and β -alanine, aminoisobutyric acid, as well as other amino acids could be found in prebiotic Earth in a much higher concentration than arginine.

I think, the other organic compounds (Quinacridone, Phthalocyanin) used in the experiments could not be found on the prebiotic Earth. Am I right? This should be pointed out in the article.

In the Introduction section, a few lines about how AMP, CMP, UMP or GMP could be produced under prebiotic chemistry conditions, could contextualize the article as prebiotic chemistry (Nature 459, 239-241, 2009. DOI:10.1038/nature08013; Geochim. Cosmochim. Acta, 265: 495-504, 2019. DOI: 10.1016/j.gca.2019.06.040).

Figure 1C: AMP, QAC and graphite were tested separately as negative controls. However, should combinations of AMP plus QAC, AMP plus graphite, QAC plus graphite be also tested?

Figure 3: The charge of these molecules depends on the pH of the medium. What was the pH of the experiments? What was the effect of these nanoenvironments on the pKas of these molecules? Could the pKas of these molecules have an effect on the covered graphite area? Probably, if these issues were better explained, a better understanding of why arginine and AMP had the largest covered graphite area could be achieved.

Finally, I want to say to the authors that I really enjoyed reading this work and the solution presented to get around the water paradox is very interesting. However, I would like to make a suggestion for future works: water activity can also be reduced by the addition of salts. Several recipes for artificial sea water are described in the literature. I believe that, from the point of view of prebiotic chemistry, it would be much more plausible to study the effect of salts than of organic molecules.

Dear reviewers,

to begin with, please note, that all the changes in the manuscript have been marked by using a blue font colour.

Comments Reviewer 1:

Figure 1 is not clear. Is the “AMP” legend the polyA oligomers <10 nt? And the AMP>10 nt refers to oligomers with more than 10 nt?

This Figure needs to be clearer, as it resumes the main results obtained in the research. The lines in gray are almost invisible but, as they show different stages in experimental design, some color variation could enhance the visibility of the results.

- ➔ **Thank you very much for this valuable comment. The legend was adjusted: “AMP > 10 nt” was changed to “AMP (Isoprop.)”, as “> 10 nt” was used to describe the precipitation method with isopropanol. We realized that this name was confusing.**

Figure 2a has too many tags in axis “y”, which looks too busy; please omit half of them to make it clearer. For instance, you could omit -0.20, 1.20, 2.60 and so forth. Do not delete the secondary marks.

Figure 2b. Same as before. Please also omit some tags on axis “y”, leave the secondary marks.

- ➔ **Thank you for this hint. The tags were reduced in both figures.**

To test the stacking ability to increase the RNA concentration, they analyzed the effect of arginine in UMP-based samples. The production in PolyU yield increased immensely. In this regard, I wonder if the authors made measurements with other nucleobases. I understand that the experiment was designed to test the feasibility to induce a larger polymerization and the selected molecule was UMP, but it could be interesting to test the other NMPs to reinforce the idea of the interaction between amino acids and nucleic acids.

- ➔ **Experiments to test the effect of arginine with other nucleobases are indeed planned for a separate project. However, to plan that additional comparative study we carefully tested if no confounding variables arise with specific combinations of samples and components. When testing GMP we observed that its precipitates are very resistive to our standard protocol procedures with regard to redissolving gel pellets. Thus, subsequent pipetting for analysing RNA concentration is problematic and produces no reliable results. This confounding variable in GMP didn't arise with precipitates of UMP or other nucleobases. Confounding variables that appear only in specific sample types are especially problematic for a comparative study where a *ceteris paribus* approach is essential. Thus, we have to understand the specifics of GMP better to adapt the protocol in a way that it can be applied unchanged for all nucleotides, enabling reliable comparison for achieving robust results for such an extended study. This study is currently part of a thesis in our group and will be published elsewhere.**

Figure 3. Please explain if the $p < 0.01$ is applicable for all the systems except for the arginine system with $p < 0.0001$. The nomenclature is not clear. The “arginine” sample is only the amino acid, or it also includes the UMP?

- ➔ A very valuable comment! As a consequence, we adapted Figure 4 in the revised manuscript (the former Figure 3) by adding lines in such a way that it becomes clear to which systems the respective p-values are related.
- ➔ The arginine sample is indeed only the amino acid. The reason is that in the context of this experiment the aim was to *separate* in a ceteris paribus approach the impact of arginine on nanofluid phenomena from the impact of other biomolecules to understand the role arginine. This is in contrast to the polymerization experiment (Fig 1a) where we tested the *interplay* of arginine with a nucleotide that was selected due to its very low self-stacking tendency.

The authors designed the experiment with other relevant inorganic materials, like magnetite and silica particles. Please specify the size of such particles and whether they are synthetic or natural solids. If they are natural, did you characterize them by means of geochemical or mineralogical methods? How did you clean the samples and select their size (and why so)?

- ➔ Thanks for the important hint. The used particles are synthetic solids with specified particle sizes and purity. In the revised manuscript we included the requested information in the materials section.

DMM calculations of the nanofluidic properties in the studied systems are appealing. The authors show that the distance between the confining surfaces is key: when the gap is reduced, the nucleotides associate and polymerize. It may be out of my own curiosity, but did you model other surfaces aside from graphite? Considering the great effect that surfaces have on polymerization, as demonstrated in your experiments, could this be the next step?

- ➔ We appreciate your comment and your curiosity on the simulations! Indeed we are currently preparing dynamic force field calculations to model other surfaces aside from graphite, especially geochemically ubiquitous metal oxides such as iron oxides and silicates, including clay minerals. However, modelling metal atom containing systems is not straightforward as canonical force field algorithms for noncovalent interactions such as DREIDING are best suited for organics, but not well suitable to metal atoms in crystals. Thus, some adaptations have to be made and implemented as additional scripts written by our collaboration partners in a separate project. Computer simulations can be a trap because they quickly produce pretty pictures. To avoid falling into “hollywood science” and ensure robust results, interaction parameters, partial charges and other items have to be defined and implemented very carefully – especially for large, complex modellings that combines the specifics of organic molecules and inorganic, metal atoms containing crystals and cluster. But we have great collaboration partner with cautious approaches so we are looking forward to the results.

Further notes

• *This manuscript is well written and logically presented; the authors made a review of the main articles published in the fields. I also recommend including this one:*

Wu, K., Chen, Z., Li, J., Li, X., Xu, J., & Dong, X. (2017). Wettability effect on nanoconfined water flow. Proceedings of the National Academy of Sciences, 114(13), 3358-3363.

<https://www.pnas.org/doi/10.1073/pnas.1612608114>

➔ **Thank you very much for recommending this valuable paper. We included it in our introduction and reference list.**

With regard to the methods, I have some comments.

Suspension preparation

L. 303. The expression “overnight” is vague, please make explicit the actual time span in hours.

➔ **We made explicit the actual time span, which was 24h.**

L. 305. The same applies to “Room Temperature (RT)”, as is not accurate, please state an actual temperature.

➔ **Thanks for the hint. We explained more precise the temperature setting of the centrifuge: Afterwards the samples were subsequently centrifuged at 14.000x g, for 1 h without cooling (room temperature between 20° C – 25°C).**

L.316. The phrasing “in an appropriate amount of H2O dest.” is vague, please explain what do you mean by appropriate amount?

➔ **Thank you very much for this comment. We explained more clearly what was meant with this phrasing: “Formed gel pellets were dried at 37°C for 20 min and resuspended in an appropriate amount of nuclease-free H2O, as little as possible needed to dissolve the gel pellet. Depending on the volume of the gel pellet, more or less volume of water was added. Final RNA concentrations were then normalized to a uniform volume of 200µl, using the total RNA amount measured and the final volume.”**

The statistics of data are robust, but the figures can be improved to make them more attractive and clearer.

➔ **We improved figures 1, 2 and 4 (the former Figure 3) in the revised manuscript.**

Comments Reviewer 2:

Critical Flaw 1): The authors state “Current strategies to circumvent this [the water] paradox have low prebiotic plausibility regarding the principle that evolution builds on existing pathways.” This logic, used to reject previous proposals for prebiotic polymerization in an aqueous environment, including the use of chemical condensing agents, alternative solvents, or wet dry cycles, suffers from the fallacy of Extended Analogy. While there are reasons to accept that current metabolic pathways have evolved from earlier pathways with similar reactions, this principle does not mean that every chemical process used in life today was exactly the same during the earliest stages of life. For example, the nucleobases were probably first produced on Earth by reactions that started with HCN, but life today does not use HCN, but amino acids, to build the nucleobases.

- ➔ **We would like to point out that the premises on which this objection is based on are not exactly those of our statement. But thanks to your comment we can clarify it: The objection that there is a logical fallacy is caused by the assumption that our statement would be based on an analogy. However, referring to the conservative nature of evolution is not drawing an analogy, it is the consideration of a basic principle that is part of the theory of evolution. If basic principles of evolution could not be applied to chemical evolution, the canonical term “chemical evolution” would not be justified. This would be an extraordinary claim that would have to be proven by those who made that claim. Another aspect is that we do not reject the mentioned previous proposals in general; we are just questioning them and do this with respect to the principle that evolution builds on existing pathways: prebiotic polymerization in non-aqueous liquids might be highly plausible to occur on worlds with dominating non-aqueous liquids such as the moon Titan. But with respect to the principle of parsimony, concepts on the emergence of life *on Earth* that are based on aqueous solutions can be regarded as those with the fewest assumptions in view on the vast evidences that early Earth was already a water world. According to this principle, such concepts can be discussed as being preferable because they tend to be more testable. They are more testable as it is harder to prevent them via ad hoc hypotheses from becoming falsified. Our statement and proof-of-concept study is in the context of just testing a hypothesis that seems to be preferable regarding the above given reasons. We made adaptations in the abstract and introduction to avoid the suggestion that we might generally reject previous proposals and also pointed out that we refer in this context to theories on the origin of life not in general, but on its emergence on Earth.**

Furthermore, the authors argue that “nanoconfinement” within enzymes and the different state of water in such restricted spaces justifies their argument that nanoconfinement was key to polymer formation on the early Earth. This too is a fallacious conclusion. The authors seem to be unaware that the free energy stored in pyrophosphate bonds, such as in ATP and GTP, are what drive polymerization in extant life. The microenvironment of an enzyme may protect against pyrophosphate/polymer hydrolysis during a polymerization reaction, but the removal of bulk water is not enough. Polymerization requires the activation energy provided by nucleotide triphosphates.

- ➔ **Thanks to the comment we can clarify some important aspects. One aspect is that we did not make a claim such as “nanoconfinement was key regarding early Earth”. We totally agree that such a claim would be bold and exaggerated in this form because no one can ever be sure what exactly happened on early Earth. In contrast, we carefully claim in our manuscript that our study suggests that nanoconfinement was key to the observed polymer formation in our particle suspensions under the described experimental conditions. We subsequently noted that particle suspensions are ubiquitous and thus among plausible geochemical settings. Another aspect is that our explanatory model does**

not suggest that nucleotide triphosphates are not important for early biochemistry. Of course we are aware of the role of ATP in biochemistry, including polymerisation. While nucleotide triphosphates very probably played an important role already in the beginning of biochemistry, the central point of our study is: nonezymatic nucleotide polymerisation can occur in experimental settings even without relying on added nucleotide triphosphates. We derived this from multiple experimental evidence and provide a thermodynamic and nanophysics based explanatory model. Our experimental findings indicate that the removal of bulk water is sufficient for inducing polymerization under the experimental conditions described in our manuscript. It can be speculated at which stage of chemical evolution nucleotide triphosphates became important and even essential for such biochemical reactions in order to run them more efficiently and controlled. But our various, independent experimental results indicate that already at a relatively early stage of chemical evolution nucleotide polymerisation can occur in an aqueous geochemical model system even without such additives.

Critical Flaw 2): 1. The authors describe their RNA polymer product isolation procedure as: "Ethanol precipitation to isolate possible poly(A) RNAs of 2-10 nt lengths and isopropanol precipitation to isolate such RNAs longer than 10 nt were applied. The results of both precipitation methods reveal that poly(A) RNA strands have formed in significant amounts regarding the negative controls (Fig. 1c) and that oligonucleotides longer than 10 nt lengths are abundant in such samples." The authors seem to be unaware of the fact that some AMP will precipitate under the conditions used to precipitate RNA polymers. The AMP that is precipitated during their described procedure may be giving a false positive reading in their fluorescence assay, and precipitation may be enhanced by their suspended particles. Furthermore, the authors also state: "The supernatant was discarded, leaving approximately 20 µl of it inside the reaction tube, in addition to any formed gel pellet." A fraction of AMP, if still suspended in ethanol solution, could result in a false positive for product formation in their assays. The authors need to show that no oligomers are present as minor contaminants in their stock supplies of AMP.

➔ **Thank you very much for this comment. It made us understand, that we need to explain more precisely the specificity of the Qubit fluorescent assay. The manufacturer of the assay kit emphasises explicitly in its description:**

"The assay is highly selective for small RNA over rRNA or large mRNA (>1,000 nt) (...), and tolerant of common contaminants such as salts, free nucleotides, solvents, detergents, or protein (...)." This declaration was controlled by measuring a 50 mM aqueous AMP solution with the Qubit® 3 Fluorometer and was validated." We added this declaration and our subsequent statement in the revised version of the manuscript This declaration and our validation means that even if AMP would also precipitate under the conditions used to precipitate RNA, it will not result in false positive reading, as the fluorophore of the Qubit assay does not interact with free nucleotides and therefore cannot give a false positive signal. Furthermore, we previously did perform negative testing of all our stock supplies. We therefore added the following explanation to the body text:

Negative controls have been performed with all substrate and organic pigment powders used in this study. However, exemplarily only those of graphite and QAC are shown here. None of those powders led to a detectable RNA concentration. Furthermore, the RNA concentration of aqueous solutions of mononucleotides were measured (not shown here) and none of these resulted in an RNA signal in the Qubit® Fluorometer. Therefore, RNA contamination of the educts can be excluded. Moreover, a negative sample using only AMP was performed, to find out if poly-A-RNA is also formed by incubation or precipitation if no substrate or organic

pigment is present. The concentration of this sample is lower than the level of RNA amount, reproducibly quantified by the manufacturer of the Qubit® Assay and can therefore be taken as lower than the reliable detection level, though it seems that very small amounts of poly(A)-RNA are also formed without adding substrates or organic pigments.

Regarding their physical assays, UV-vis melting temperature is used as an assay for poly(A) formation. However, poly(A) does not give a 2-state melt as it would in the presence of a poly(T) complement. Under acidic conditions a hemi-protonated poly(A) duplex is formed, but there is no indication that their conditions would support the formation of this highly pH-dependent structure. Results from their assay shown in Figure 2 reveals that the negative control sample also gives a melting transition above 80°C. The origin of this transition in the control sample is equally perplexing.

➔ **Thank you for this comment. We realized, that the used method was not clearly explained and we therefore added to the methods section the following:**

“To obtain more information about the length of the formed RNA we performed a comparative melting curve analysis after the last cycle of the qPCR.”

Please note, that the melting curve is not a UV/Vis spectrum but is performed by the Light-Cycler: The instrument heats up the samples to 95°C, while constantly measuring the fluorescence level of the amplicons, formed by previous PCR. By heating up the samples, the hydrogen bonds of the DNA double strands (PCR products) break, leading to single stranded DNA. One can roughly say, the longer a DNA piece the higher the melting temperature needs to be, to break all hydrogen bonds. However, the composition of the DNA sequence also influences the melting temperature. The fluorophore is designed in a way that it only emits a fluorescence signal when it is bound to DNA double strands. In the moment all double strands are split into single strands, the slope of the fluorescence as function of the temperature becomes zero and the derivation $-(d/dT)$ Fluorescence shows its maximum. Which is shown in figure 2.

Please note, that Figure 2 does show the results of the quantitative PCR (qPCR), including its quantification over the cycles and the melting curve (explained above), as written, and not a UV/vis spectra, which we did not perform.

The negative sample is yellow in Figure 2a and does not show an increase of fluorescence, as a consequence that no DNA amplicons are formed here, demonstrating that there is no amplifiable cDNA (formed by previous reverse transcription) in this sample.

In Figure 2b the tested sample is grey and the positive sample is green (as written and explained).

The authors use Quantitative PCR after reverse transcription (RT-qPCR) to confirm the synthesis of poly(A) from AMP. This technique is highly prone to false positives due to contamination. It is known that some commercial sources of AMP have poly(A) contaminants. The processes/additives used as part of the RT-qPCR experiments, including particles in suspension, could be facilitating the amplification of such contaminants.

➔ **As already mentioned before, all necessary negative samples were performed on all used educts, so that we can exclude any contamination. Furthermore a negative sample within the run, starting at reverse transcription (as written in methods section) shows that neither any educt of the RT-qPCR was contaminated nor the cycling plate nor other material used to perform RT-qPCR.**

I recommend that authors completely redesign their polymerization assay to include more standard analytical techniques, such as gel electrophoresis, HPLC, and mass spectrometry.

- ➔ **Following this recommendation, we performed *capillary gel electrophoresis* as an additional, independent standard analytical technique. The results were included as Fig. 3 in our revised manuscript and a the discussion and methods part of the paper was extended accordingly.**

Comments Reviewer 3:

I have few questions that I would like that the authors clarified them.

In the title of the article appears the word “prebiotic” this means that the experiments were performed under reaction conditions that could have existed on the prebiotic Earth.

- ➔ **Thank you for this comment! It helps us to reflect on the title and check if the adjective is properly used. There are indeed two possible meanings of the term “prebiotic” according to the *American Heritage Dictionary of the English Language, 5th Edition*: (1) relating to the conditions prevailing on earth before the appearance of living things; (2) occurring before the advent of life. In our title the adjective “prebiotic” relates to the second definition (in the sense of “nonenzymatic polymerisation without biochemistry” – before the appearance of living entities).**

I agree with the authors that magnetite and silica were widespread minerals on the prebiotic Earth. However, arginine and anthraquinone were not widespread organic compounds on the prebiotic Earth. According Milshteyn et al. (2019) cited by the authors, the anthraquinone concentration in the Murchison meteorite was 2.0 nmoles/g (Sci. Rep. 2019, 9, 12447. DOI: 10.1038/s41598-019-48328-5). This means that the concentration of anthraquinone in the primitive seas of the Earth was very low. Also, arginine is one of the amino acids that were only synthesized in few experiments and always very low amount (Orig. Life Evol. Biosph. 2008, 38: 469–488. DOI 10.1007/s11084-008-9150-5). This issue should be addressed by the authors.

- ➔ **We fully agree that the total amount of anthraquinone would be low under the assumption that it only originated from exogeneous sources (meteorites). But that this assumption does not include photochemical processes in the upper atmosphere of early Earth. It might be relevant to include such processes in this discussion as they probably have produced an organic haze that contains a vast amount and variety of polycyclic aromatic hydrocarbons (PAHs) similar to present-day moon Titan according to prebiotic models of early Earths atmosphere (see e.g. *Curr. Org. Chem.* 2013, 17, 1710. DOI: 10.2174/13852728113179990078). If this haze would have been global as it is on Titan, even a huge amount of PHAs and their photochemically produced heterocyclic derivatives such as anthraquinone would have been present as insoluble particles within the aerosol droplets and – via rain – in the primitive seas. Although this is based on current models on prebiotic atmospheric syntheses, its is speculative, so we didn't include that in the present manuscript as it would distract the discussion from the main target of the article. As a consequence, we focused on just citing the non-speculative occurrence of prebiotic anthraquinone in meteorites. However, inspired by your comment, we added to mention the geochemical setting of watery microparticle suspensions in the form of aerosols with particle burden and the prebiotic relevance of aerosols in the revised manuscript, including respective references. Another aspect that could be relevant in this context is the fact that anthraquinone is an organic semiconductor. Semiconducting quinones play a crucial part in biochemistry. They are electron acceptors and thus essential in electron transport chains (e.g. in photosynthesis). This electrochemical quality may outcompete possible low quantity. The same with arginine: its outstanding stacking affinity with other aromatics such as nucleobases is a very important quality that may have been more important on early Earth (e.g. for catalysing condensation of nucleotides) than quantity.**

As a suggestion for other works of the group. Could other amino acids be used for these experiments? Glycine, α and β -alanine, aminoisobutyric acid, as well as other amino acids could be found in prebiotic Earth in a much higher concentration than arginine.

➔ **Thank you very much for your valuable suggestion! The selection criterion that leads us to arginine was its outstanding stacking tendency with aromatic groups. In the context of our study this property was key to test our hypothesis on the role of stacking. But, yes: other amino acids could be used as well, we just have to select some regarding the limited time and funding and thus have to find a reasonable selection criterion for a thesis we are planning. Therefore, we are very thankful for your suggestion and list of some amino acids that probably occur in higher concentration on early Earth, indicating this as another valuable selection criterion for other studies.**

I think, the other organic compounds (Quinacridone, Phthalocyanin) used in the experiments could not be found on the prebiotic Earth. Am I right? This should be pointed out in the article.

➔ **Phthalocyanine is a model system for the class of porphyrines that have key roles in biochemistry (e.g. photosynthesis, oxygen transport) and are discussed as prebiotic relevant compounds that could have been found on early Earth under conditions on primordial volcanic islands. Due to your comment, we added this information in the supporting materials section and include references both in the main paper and supporting materials. So far, there are no publications on Quinacridone that claim direct prebiotic importance. As pointed out in our article, the main reason we selected Quinacridone (and graphite) is the fact that it is an intensively investigated system in the context of inducing nanofluid phenomena in aqueous particle suspensions. Due to this fact we were able to base our claims on a system that is well understood in this context. This enabled us to avoid unexpected confounding variables and thus becoming able to focus on the biomolecules in our ceteris paribus approach and controlling their variables. However, we want to note that quinacridone is a polyaromatic heterocycle. As mentioned above, models on atmospheric conditions on early Earth suggest the presence of an organic haze layer consisting of polycyclic aromatic hydrocarbons (PAHs) and heterocyclic derivatives as products of photochemical reactions. Quinacridone is a heterocyclic derivative of the PAH pentacene (as anthraquinone is a derivative of the PAH anthracene). So it is quite possible that it could have been found on early Earth in organic haze (and thus also in the oceans, transported via rain) as one of the photochemical products. If an atmospheric haze would have been global as in the case of present-day Titan, the amount of this heteroaromatic compounds could even have been very high on early Earth.**

In the Introduction section, a few lines about how AMP, CMP, UMP or GMP could be produced under prebiotic chemistry conditions, could contextualize the article as prebiotic chemistry (Nature 459, 239-241, 2009. DOI:10.1038/nature08013; Geochim. Cosmochim. Acta, 265: 495-504, 2019. DOI: 10.1016/j.gca.2019.06.040).

➔ **Thank you for this constructive comment. We included a sentence on promising prebiotic synthesis strategies for nucleosides and nucleotides and added references on recent review articles on that topic.**

Figure 1C: AMP, QAC and graphite were tested separately as negative controls. However, should combinations of AMP plus QAC, AMP plus graphite, QAC plus graphite be also tested?

➔ We indeed also tested AMP + QAC (0.57 ng/μl) and AMP + graphite (1.37 ng/μl) as analysis samples, but did not show the data of all tested analysis samples as this would far exceed the discussion of this contribution (they will be published elsewhere). Regarding negative controls: We did not test the combination QAC + graphite as additional negative control, as we separately tested both compounds as negative samples and without the addition of nucleotides, formation of RNA in a combination of both did not seem plausible to us before this background.

Figure 3: The charge of these molecules depends on the pH of the medium. What was the pH of the experiments? What was the effect of these nanoenvironments on the pKas of these molecules? Could the pKas of these molecules have an effect on the covered graphite area? Probably, if these issues were better explained, a better understanding of why arginine and AMP had the largest covered graphite area could be achieved.

➔ Thank you for your constructive questions and suggestion. We added pH values to the respective section of the revised manuscript. Indeed, higher pH also led in almost all our samples to higher coverage in the same extent. A higher pH shifts the zeta potential to a more negative one, which in turn increases the surface coverage of graphite. Interactions due to correlations in charge fluctuations and the effects of induced polarization charges at dielectric discontinuities contribute to increased attractive forces between nanoscale separated particles and surfaces in vicinal water and are an important factor that increases surface coverages. However, this point is not explained in detail in this manuscript, as this would far exceed the scope of this publication. But it was the focus of one of our previous publications (*JACS* 140 (4), 1327–1336 (2018)) to which we referred in our current manuscript. In this previous publication we noted that AMP-containing samples are not quite following the pH/zeta potential trend regarding surface coverage. Our current hypothesis is that the high self-stacking tendency of AMP (and arginine) leads to the formation of a highly correlated liquid within the nanoconfinements – an effect that contributes to the emergence of Casimir-like fluctuation induced forces. However, this hypothesis still needs to be refined and its discussion would far exceed the scope of this submitted contribution (but will be published elsewhere). So, we decided to come to the very point in the discussion of this submission via the following consideration: Casimir-like forces arising by fluctuations of general order parameter (e.g. correlated charge and order fluctuations) are part of the complex network of nanofluid effects in confined water. OSWD emerges from such effects. So we focused on OSWD to investigate and discuss synergy between nanofluid phenomena (influenced e.g. by pH and stacking tendency) and nucleotide polymerisation. This allows to condense the complexity of the discussion and enables us to keep the target of this paper in focus. In the revised version we marked in colour the reference to our previous study that explains these issues.

Sincerely Yours,
Frank Trixler

REVIEWERS' COMMENTS:

Reviewer #1 (Remarks to the Author):

Dear authors and editors,

After reading the latest and improved version of this article, all possible doubts that I raised in my previous review have been cleared up. The article is clearer and presents more evidence (due to the use of other methods) that gives more robustness to its approach.

I think this article is extremely interesting and, as I said before, it contains very relevant data that significantly contributes to the discussion on the subject.

I consider that in its current state the article can be accepted and published.

Sincerely,

Reviewer #3 (Remarks to the Author):

Dear Dr. Trixler

Good afternoon, In my opinion, your answers to my questions have solved most of my doubts. Therefore, I recommend publishing the article. I would like to take this opportunity to say that your work has very interesting data for prebiotic chemistry. I am curious to know how your system would work in the presence of artificial seawater. I hope one day to see some of your group's work in this condition.